# Low Recombination Firing-Through Al Paste for N-Type Solar Cell with Boron Emitter

**DOI:** 10.3390/ma14040765

**Published:** 2021-02-06

**Authors:** Peng Zhu, Yuan Liu, Chengjiang Cao, Juan Tian, Aichuang Zhang, Deliang Wang

**Affiliations:** 1College of Chemistry and Chemical Engineering, Nantong University, Nantong 226019, China; pzhu@ntu.edu.cn (P.Z.); 1908320010@stmail.ntu.edu.cn (C.C.); 2007320013@stmail.ntu.edu.cn (A.Z.); 2Jiangsu Province Cultivation Base for State Key Laboratory of Photovoltaic Science and Technology, Changzhou University, Changzhou 213164, China; 3Hefei National Laboratory for Physical Sciences at Microscale, University of Science and Technology of China, Hefei 230026, China; tian999@mail.ustc.edu.cn

**Keywords:** low recombination, fire-through, aluminum paste, N-type solar cell, boron emitter

## Abstract

A kind of low recombination firing-through screen-printing aluminum (Al) paste is proposed in this work to be used for a boron-diffused N-type solar cell front side metallization. A front side fire-through contact (FTC) approach has been carried out for the formation of local contacts for a front surface passivated solar cell. With a low contact resistivity (ρ_c_) of 1.0 mΩ·cm^2^, good ohmic contact between the boron-doped front surface of the silicon sample and the Al paste was realized. To obtain a good energy conversion efficiency, a balance can be achieved between the open circuit voltage (V_oc_) and contact resistivity (ρ_c_) of the cell by combining suitable Al powders and appropriate additives. The detailed micro-contact difference in Si/metallization between the firing-through Al paste and silver-aluminum (Ag-Al) paste was analyzed. The dark saturation current density beneath the metal contact (J_0, metal_) of the Si/metallization region using our firing-through Al paste was discussed, which was proven to be 61% lower than using Ag-Al paste. The pseudo energy conversion efficiency of the cell using Al paste measured by Suns-V_OC_ was also higher than using Ag-Al paste. The role of Al paste in low surface metal recombination is discussed. The utilization of this new kind of Al paste was much cheaper and more convenient, compared to the traditional process using Ag or Ag-Al paste.

## 1. Introduction

Over recent decades, silicon (Si) solar cells have been widely used in various lighting and power generation systems. Electric power using solar cells through photoelectric conversion from solar energy can be obtained as a pollution-free and high supply resource. A passivated emitter and rear cell (PERC) based on P-type Czochralski-grown silicon (Cz-Si) substrate has been introduced into production [1]. However, with absence light induced degradation (LID), higher lifetimes of minority charge carriers, and lower sensitivity towards metal impurities, N-type solar cells could offer greater efficiency possibilities and advantages over P-type solar cells [2,3,4,5,6,7].

There is no doubt that the metallization process forming the electrode has turned out to be an important part of the enhancement of energy conversion efficiency, which adopts screen-printing paste composed of metal particles, low-melting glass frits and an organic part. Nowadays, silver (Ag) pastes applicable for the metallization of bifacial N-type solar cells through screen-printing technology are commonly investigated by many researchers [8,9,10,11,12,13,14,15,16,17]. For the formation of ohmic contact on the boron-doped emitter, since the mechanism is more complicated, many attempts have been made to improve the semiconductor-metal contact. Some reports have proved that the contact resistivity between the electrode and the Si substrate could be remarkably reduced by adding aluminum particle to the Ag paste to form silver-aluminum (Ag-Al) paste [18,19,20,21,22,23,24]. On the other hand, with the presence of the Al particle, some inappropriate Ag/Al spikes can give birth to a larger dark saturation current beneath the paste contact area [25,26,27]. Lago’s group proved that the spiking problem could be solved by adding silicon powder to the metallization paste [23]. Subsequently, Lohmülle et al. suggested that increasing the junction depth (deeper than 1 μm) or the formation of Ag nanocrystals can reduce the recombination loss beneath the Ag-Al contact [28]. In 2016, it was reported by Kumar et al. that 20% efficiency had been reached using Ag-Al paste [21].

However, the large-scale application of N-type silicon solar cells has also been limited by the cost of the wafer and bifacial Ag paste. It has been an urgent issue to cut down the cost of N-type silicon solar cells. In particular, aluminum conductive paste has become an important material for the back side metallization in the formation of the back surface field (BSF) in P-type silicon solar cells [29,30,31]. In this case, the charge carrier’s lifetime could be significantly prolonged. Light transmission and the charge carrier’s neutralization could also be reduced. Good performance of the solar cell could be achieved with this cost-effective option. However, rather than forming regenerative nanoparticles, aluminum electrodes on a boron-doped emitter of ohmic contact should be more difficult. Few reports on Al paste being applicable for the front side metallization of N-type solar cells have been published up to now. On the basis of the expected performance of Al paste with low surface recombination of the wafers [32], Al conductive paste should be considered as an alternative anode conductive material for the front side boron emitter metallization of N-type silicon solar cells. The utilization of an Al electrode on the boron-doped emitter could be acquired at both high photoelectric conversion efficiency and low cost.

There are three approaches for the formation of local ohmic contact in front surface passivated solar cells: chemical structuring, laser process and fire-through contact (FTC) [33]. Usually, the creation of local contacts in the fabrication of PERC solar cells can be realized by a chemical structuring or laser process resulting in local ohmic contacts during co-firing, which is prone to seriously damage the passivation layer. A fire-through contact approach, employing a screen printing process and leading to no damage of the passivation layer before firing, contribute to the integration of bifacial passivation concepts in traditional production lines.

In this work, a type of firing-through Al screen-printing paste applied on the boron-doped emitter N-type solar cell is presented. The fire-through contact approach for the N-type passivated emitter and rear totally diffused (N-PERT) solar cell with a screen-printed approach is evaluated. The investigation of the contact mechanism between the Al paste and Si wafer has been carried out through micro-structure analysis at the Si/metallization interface. Both the Al powder and the additive of the Al paste were checked to understand the formation process of ohmic contact on the front side metallization for the silicon solar cell. Based on this, the passivation layer could be etched away with Al paste. Afterwards, ohmic contact with the Si substrate could be formed by optimizing the constituents of the paste. Good ohmic contact, between the boron-doped front surface of the silicon wafer and the Al paste, for a front electrode was realized, with low contact resistivity ρc at 1.0 mΩ·cm^2^. To obtain good energy conversion efficiency, a balance can be achieved between the open circuit voltage (V_OC_) and contact resistivity (ρ_c_) by adding appropriate additives. The role of the Al paste in low surface recombination is also discussed. The experimentally extracted dark saturation current density beneath the metal contact (J_0, metal_) in the front side of the silicon cell using our aluminum paste was determined by Suns-Voc, and was proven to be 61% lower. The energy conversion efficiency is 0.4% higher than that of Ag-Al paste. Compared with the traditional process by using Ag or Ag-Al paste for the boron-doped emitter, it is much cheaper and more convenient using Al paste, which is instructive and significant for the current transmission mechanism of Si/metallization and improved application of Al conduction pastes.

## 2. Materials and Methods

In this study, 156 × 156 mm^2^ N-type Cz silicon wafers with base doping of 1 Ω·cm were used. The p^+^ (boron) diffused regions with a thickness of 0.5 μm were formed with a sheet resistance of 90 Ω/sq. at the front side of n-type Si substrate. The n^+^ (phosphorous) diffused regions with a thickness of 120 nm were formed with a sheet resistance of 25 Ω/sq. at the back side. This diffused region was passivated by a 1–2 nm Al_2_O_3_ layer covered with a 70 nm anti-reflection SiN_x_ layer on the sample exterior surface. The SiNx and Al_2_O_3_ layers were prepared by plasma-enhanced chemical vapor deposition (PECVD). The manufacturing process and the structure diagram of the N type passivated emitter and rear totally-diffused (N-PERT) solar cell are shown in Figure 1.

The firing-through Al paste used in this paper consists of Al powder (74 wt%), glass frit (5 wt%) and an organic part (21 wt%). The ingredients above were mixed together then made into paste through centrifugal dispersion and three-roll grinder. The Ag-Al paste used on the front side of cell as a comparison was also a present state-of-the-art product for the N-type solar cell. A commercial silver (Ag) paste (Nantong T-Sun New energy Co., Ltd., Nantong, China) was used for the rear side metallization of the cell. The metallization process on both sides of the wafers was employed by screen-printing metal pastes with an H-pattern grid design undergoing the screen printing process. The screen printer was purchased from Baccini (Treviso, Italy). Afterwards the samples were fired in a belt sintering furnace with a peak temperature of 750–760 °C.

An image of a completed solar cell is shown in Figure 2, which has 5 bus bars and 106 fingers of metal paste lines. In this paper, the electrical characterizations of the solar cell as in Figure 2 were analyzed with light under standard AM1.5 illumination (1 kW/m^2^, 25 °C) and dark IV measurements on a solar J-V tester (IVT Solar Pte Ltd., Shanghai, China) [34]. Besides, the implied electrical properties of the cell were measured by Suns-Voc (Boulder, CO, USA). A transfer length model (TLM) method on 10-mm-width stripes with equidistant metallization lines has been applied to acquire the contact resistivity (ρ_c_) [35]. 

To obtain the recombination current density beneath the metal contact (J_0, metal_), different front side screen-printing patterns with metallization fractions of 3%, 6%, 9% and 12% have been designed by changing the number of the screen-printed fingers, as shown in Figure 3. The measured J_01_ by Suns-V_OC_ can represent the recombination of the entire cell when the ideality is about 1 [36]. According to [36,37,38], the slope of the linear regression of the J_01_ vs. metallization fraction allows determination of the (J_0, metal_ − J_0, pass_). The recombination current density (J_0, pass_) of emitter passivation, known from experimentally fully passivated areas, is around 25 fA/cm^2^. 

The detailed micro-structure analysis at the cell surface region has also been developed by a field-emission scanning electron microscope and energy dispersive spectroscopy (EDS) (FE-SEM, Sirion 200, HITACHI, Tokyo, Japan). The SEM of the Si substrate surface was taken after soaking the cell in 60% nitric acid for 10 min to remove the paste.

## 3. Results

### 3.1. The Mechanism of Ohmic Contact between Al Paste and Si Substrate

N-PERT solar cells using Ag-Al paste and firing-through Al paste for boron-doped emitter metallization are denoted as Cell 1 and Cell 2, respectively. The distribution of measured contact resistivity (ρ_c_) and open circuit voltage (V_OC_) of the investigated solar cell with Ag-Al paste and Al paste are shown in Figure 4, and the average value is displayed above each box plot. To our knowledge, it is the first result of a solar cell proving that Al paste could be applied at the front side boron emitter in the N-type solar cell. Compared with Ag-Al paste, slightly higher ρ_c_ through Al metallization was yielded, while higher V_OC_ were obtained. This indicates that the firing-through Al paste is a promising candidate for the metallization of the boron emitter for improved performance of N-type silicon solar cells [36]. 

The J_01_ values of the small solar cells metallized with Ag-Al paste and firing-through Al paste using the printed pattern mentioned above are shown in Figure 5. The solar cell metallized with the firing-through Al paste yield as significantly lower J_0, metal_ compared with that of the cell using the Ag-Al paste. This indicates that the firing-through Al paste is a promising candidate for the metallization of the boron emitter to obtain a lower metal recombination and improved performance of N-type silicon solar cells [36].

The dark J-V curves of Cell 1 and Cell 2 shown in Figure 6 demonstrate that the Al paste makes a contribution to a lower current leakage current density when applying a backward voltage, which implies much fewer defects in the Cell 2 resulting from Al paste. At high forward voltages, the two dark J-V curves demonstrate a typical exponential current increase with the voltage, indicating low Rs of the two cells, which is consistent with the contact resistivity results shown in Figure 4. To confirm the performance of metallization, more work including analysis of microstructure and component influences is needed to investigate the mechanism of ohmic contact between the metal and Si substrate.

The microstructural properties of the Si/metallization interface are examined by SEM at 10 kV. SEM results confirmed that the damage to the Si wafer was caused by metal paste. The typical Ag-Al paste/Si substrate microstructure is shown in Figure 7a. The Ag-bulk/Si contact in p+ emitter has been studied in detail [9,11,12,16], and the glass frit can etch away the passivation layer through firing-through contact technology. As reported [13], the glassy-phase plays an important role in contact properties, in which regenerated nano-Ag particles are dissolved to form Ag colloids. The Ag colloids with the glassy-phase assist the current tunneling with the intention of realizing current transmission through the glass phase. The SEM image of the Si/metallization interface after the metal paste was removed is shown in Figure 7c. A continuous phase containing Ag colloids was observed at this region, which is the same as reported: that the combination influences of dissolved Ag particles and glassy-phase have crucial effects on the current transport across the interface [21].

However, the SEM micrographs in Figure 7b show that the glass frit did not facilitate the precipitate containing Al particles in the contact region, which was also proven in the Si/metallization interface in Figure 7d. The optimally fired-through contact was implemented to clarify the pattern of current transmission between the Al paste and the Si substrate. We suggest that the current transmission between the firing-through Al paste and the Si substrate was formed by a direct contact and a current tunneling between Al bulk and the silicon substrate through the 1–2 nm thin Al_2_O_3_ layer. It is also interesting that, without the continuous glassy-phase on the surface of Si substrate, less etched region was found in the Al/Si substrate interface. The reserved area of passivation layer shows no obvious damage, leading to reserved passivation.

EDS analyses were also employed to examine the reaction between the Al paste and the passivation layer (SiN_x_). The cross-sectional microstructures of the Si/metallization region without an Al particle directly attached are shown in Figure 8a. The element mapping images of this region are shown in Figure 8b–d correspondingly. According to a previous study on the sintering mechanism of Ag paste [10,11,12,13,14,15], these “voids” in the passivation stack area (Figure 8a) are formed by the N_2_ produced from the reaction between the glass frit and SiN_x_, given the composition of glass frit used in our Al paste. The element mapping results also confirm that no O element is contained in the “voids” area. In Figure 8b,d, the existence of the continuous O layer and Al layer proved that the Al_2_O_3_ layer was kept without damage. Besides, it is illustrated in Figure 8c that, devoid of the continuous glass phase, when dissolving nano metal colloids on the surface of the Si wafer, the SiN_x_ layer was partially reserved. The remaining SiN_x_ layer could perform surface passivation, which gives rise to the low metal recombination of the solar cell.

Unlike the semiconductor-metal contact through the nano-Ag particle dissolved in the glass phase, the contact between Al paste and the Si substrate was produced in default of the assistance of the regenerative nano metal colloids. Those Si/metallization contact regions without an Al particle directly attached cannot collect current, as the SiN_x_ layer exists. The contact mechanism between Al paste and Si substrate has to be considered under the influence of comparatively large-scale Al particles. High and low magnification SEM images in Figure 9a,b were used to further characterize the micro-contact Si/metallization interface. The Al particles maintained a good spherical shape during the firing process resulting in a discontinuous contact with the Si substrate. Those areas where the Al particles in direct contact with the Si substrate could collect current from the Si substrate are shown sin Figure 9b. Because of the existence of gaps between Al particles, the current cannot be collected through some areas on the Si substrate untouched by Al particles. As a result, some obvious etching areas occurred as shown in Figure 7d, which engendered a serious recombination at these interfaces. In other words, due to the morphological characteristics of Al particles, the reaction between the Al paste and the Si substrate was very uneven.

According to the above analyses, the high J_0, metal_ of the cell using Ag-Al paste could be attributed to the continuous Ag nano colloids on the Si substrate. For the cell using firing-through Al paste, some of the Si substrates touched by the Al particle reacted with Al violently, forming an inverted pyramid-shaped etched area on the passivation layer, generating a high surface recombination during the current transmission process. The other region without etching maintains good passivation characteristics but cannot collect current from the Si substrate at the same time. Such inhomogeneous contact does not benefit the obtaining of a high efficiency cell. On the other hand, a shallow and homogeneous etching area is necessary for the cell to get a low J_0, metal_ and ρ_c_.

### 3.2. The Effect of Silicon Particle in Aluminum Paste on Ohmic Contact

N-PERT solar cells using firing-through Al paste with and without adding Si powder for boron-doped emitter metallization are executed, denoted as Cell 2 and 3, respectively. As reported, the Si in the Al paste suppressed the driving force for the strong lateral silicon diffusion towards the Al layer [39,40,41], initiating low and weak reactivity of the paste. The distribution of measured ρ_c_ and V_OC_ of the investigated solar cells with different Al paste are shown in Figure 10, and the average value is displayed above each box plot. The V_OC_ of Cell 3 was 8 mV higher than Cell 2’, but cells’ ρ_c_ were roughly the same, which proves that adding silicon particles to the Al paste can effectively reduce the solar cell defects. 

The two cells’ dark J-V curves are shown in Figure 11 to confirm if adding silicon particles can reduce the defects of the cell or not. The leakage current density at the voltage of −1 V of Cell 3 is about one order of magnitude smaller than Cell 2, indicating fewer defects in the cell using the Al paste adding Si particles. Cell 3 exhibits better diode properties. In conclusion, we can speculate that the reason for the low reverse leakage of cell 3 is the addition of silicon particles, which reduces the defects caused by the paste on the cell.

The micro-contact morphology of the Si substrate at the Si/metallization interface has been studied by SEM. Figure 12 shows that the etching areas are smaller and shallower when adding Si particles into the Al paste, which reveals that adding Si particles into the Al paste can prevent the violent reaction between the paste and SiN_x_ layer. We can conclude that the Si particle additives can reduce the etching area significantly since the addition of Si particles can reduce the diffusion of silicon from the substrate into the paste.

### 3.3. The Effect of Aluminum Particle Size on the Cell Electrical Properties

Since the contact pattern of Al paste and the boron-doped emitter derived from joints between Al particles and the pyramid surface texture of the Si substrate, the contact and etching area depends on the size of Al particles, which play an important part in contact resistivity. In addition, the different interspaces among Al particles would be involved in the electrical properties of the silicon cell. Silicon solar cells named Cell 2, Cell 4 and Cell 5 were metalized by Al paste with different Al particle size: 1–3 μm, 5–6 μm, 9–11 μm, respectively. The distribution of measured ρ_c_ and V_OC_ of the investigated solar cells with different Al paste were shown in Figure 13, the average value displayed above each box plot. It is obvious that the Voc of Cell 5 using an Al particle with 9–11 μm particle size was 30 mV lower than that of Cell 2 using 1–3 μm particle size Al powder. When different particle sizes react with the Si substrate, the larger particle size Al powder will damage the substrate more seriously, which will cause the Cell’s V_OC_ to decrease. 

The cells’ dark J-V curves are also plotted in Figure 14. It shows a larger reverse leakage current and lower rectification ratio of the cell when using larger particle size Al powders, indicating an increment of interfacial recombination and a reduction in the shunt resistance in the cell after the metallization process, consistent with the Voc parameters shown in Figure 13.

The Si/metallization interface and the etched region of Si substrate surface after the paste removal was measured by SEM. As shown in Figure 15, the etched area of the Si substrate is obviously enhanced, with Al powder particle size increasing. Figure 15c,f show the Si/metallization contact area SEM images of Cell 5. As we can see, the etched area of the Si substrate looks like a deep pit, which was caused by the reaction between the large-size Al particles and the Si substrate. Even the p-n junction of the Cell 5 may be destroyed which is only 0.5 μm away from the front surface, as the V_OC_ of Cell 5 was so low. On the other hand, the Si substrate of Cell 2 in direct contact with the Al particles is more homogeneous in Figure 15a. These direct contact areas allow current to be transmitted, so the contact resistivity of Cell 2 is not particularly high, at 1.3 mΩ·cm^2^. The etching area caused by Al particles on the Si substrate is so shallow that a high Voc of Cell 2. We can conclude that a more homogeneous etched area of Si substrate at the Si/metallization contact interface could be achieved by using smaller particle size Al powder, as this homogeneous etched Si substrate is more conducive to balancing the Cell current transmission and recombination.

Based on the above, it is considered that the semiconductor-metal contact takes place through the glass frit attached to the Al particle reacting with the passivation layer (SiN_x_), associated with the size of the etched area. It is suggested that using smaller particle size powder is the better choice to reduce the deep etch area and obtain uniform contact to balance the contact resistivity and surface recombination.

### 3.4. The Electrical Characteristics and Micro-Contact of Solar Cells with Silver Aluminum Paste and Improved Firing-Through Aluminum Paste on the N-Pert Solar Cell

N-PERT solar cells using improved firing-through Al paste on the front boron emitter were carried out, named Cell 6. The improved Al paste was mainly prepared by adding appropriate content of Si powder and using Al powder with a particle size of 1–2 μm. The electrical characterization of the cells under 1 sun light illumination is listed in Table 1. The Jsc and FF are the abbreviations of saturation recobination current densities and fill factor. Cell 6 using improved firing-through Al paste shows a higher open circuit voltage (V_OC_), but a higher series resistance (R_S_) leading to a loss of actual efficiency (Eff). The R_S_ difference of the two cells is mainly related to the front paste fingers’ resistance and the contact resistance between the paste and the Si substrate as the two cells have used the same Si wafer and rear paste. The higher series resistance of Cell 6 than Cell 1 was mainly due to the line resistivity of Al finger (4 × 10^−5^ Ω·cm), higher than the Ag-Al finger (4 × 10^−6^ Ω·cm). 

The implied electrical characteristics of Cell 1 and Cell 6 measured by Suns-V_OC_ are shown in Table 2. The implied V_OC_ (i-Voc), pseudo fill factor (pFF), pseudo efficiency (pEff) and the first diode saturation recobination current densities (J_01_) was 655 mV, 83.48%, 21.34%, 347 fA/cm^2^, respectively. It demonstrates that a great improvement in cell implied electrical performance is achieved using low recombination Al paste comparing with Ag-Al paste.

The distribution of measured ρ_c_ and V_OC_ of Cell 1 and Cell 6 with Ag-Al paste and improved firing-through Al paste are shown in Figure 16, and the average value is displayed above each box plot. Low ρ_c_ and high V_OC_ were obtained using the improved firing-through Al paste.

Figure 17 shows the J_01_ of the small solar cells metallized with improved firing-through Al paste using the printed pattern as in Figure 3. Low metal recombination J_0, metal_ of 348 fA/cm^2^ was obtained as shown in Figure 17. 

Figure 18 shows the SEM images of Cell 6 obtained at the Si/metallization interface after the paste removal. This reveals that the homogeneous and shallow etching area on the Si substrate induces a low metal recombination and low contact resistivity. We are trying to reduce the series resistance of the cell using this firing-through Al paste, which can be achieved by designing dense grid lines and printing silver paste on the firing-through Al paste.

## 4. Conclusions

In this work, a kind of low recombination firing-through aluminum screen-printing paste on the front side for the metallization of silicon solar cell was presented. The fire-through contact approach for the N-PERT with screen printed contacts was evaluated. The electrical properties and the analysis of Si/metallization micro-structure were investigated in detail. 

Microstructure analysis indicated that a shallow and homogeneous etching area of Si substrate was necessary for balancing the Si/metallization interface recombination and contact resistivity. Adding Si particles to the paste can conspicuously attenuate the reaction between the paste and the Si substrate, in order to achieve low recombination of the reduced cell and corresponding low contact resistivity. In addition, the utility of small particle size Al powders can make the paste and the silicon substrate contact more homogeneous. The test results evince that small Al particles cause no extra deep etching area to Si substrate, which accounts for the lower recombination and suitable contact resistivity of cells. 

The results of this work make it clear that high open circuit voltage and low contact resistivity could be realized by using our Al paste. Outstanding electrical properties could be obtained. The lower J_0, metal_ of 348 fA/cm^2^ (890 fA/cm^2^ for Ag-Al paste) and higher pseudo-efficiency of 21.34% when using the Al paste for front side metallization were obtained compared with that of cells using Ag-Al paste. Although the non-optimal ρ_c_ of the fire-through contacts has been reported earlier, a low ρ_c_ value for the N-type silicon cell could be realized via our work. 

## Figures and Tables

**Figure 1 materials-14-00765-f001:**
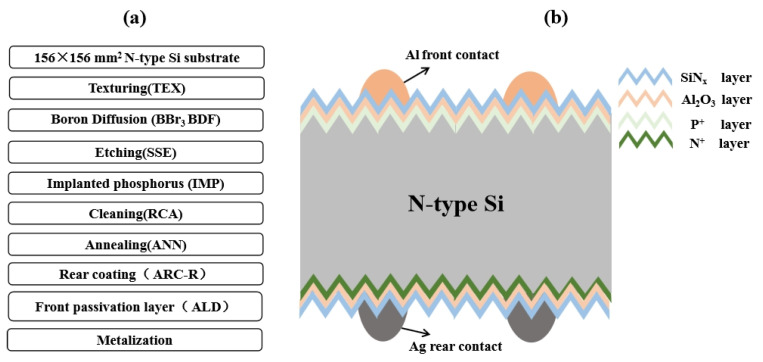
(**a**) The manufacturing process; (**b**) the structure diagram of N-PERT solar cells.

**Figure 2 materials-14-00765-f002:**
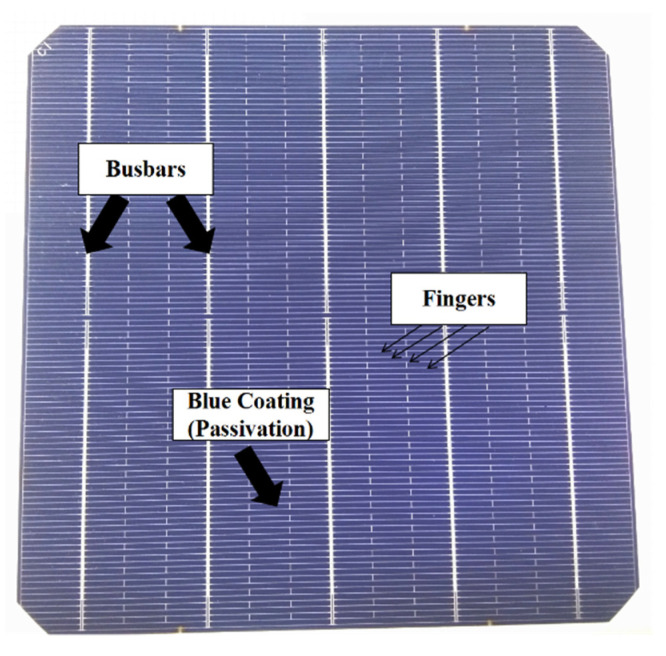
Image of a completed solar cell.

**Figure 3 materials-14-00765-f003:**
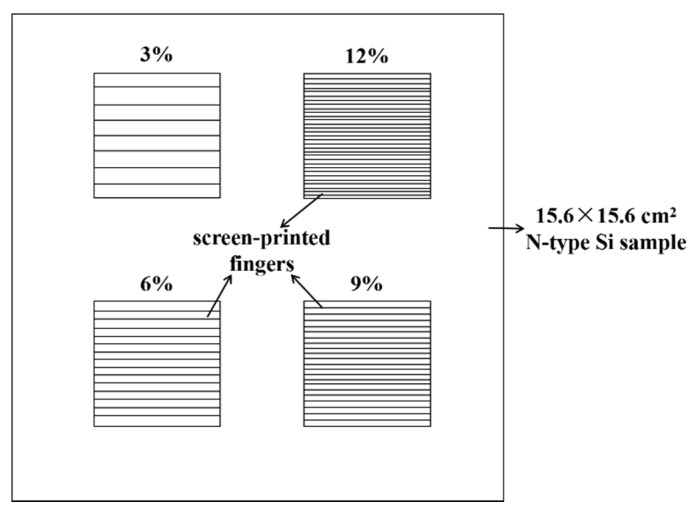
The screen-printing pattern of the sample with different metallization fraction from 3% to 12%.

**Figure 4 materials-14-00765-f004:**
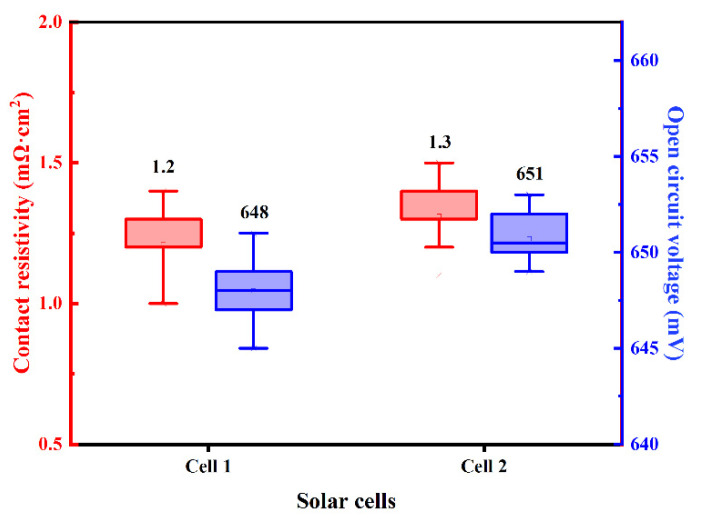
The distribution of the ρ_c_ and V_OC_ of the solar cells using Ag-Al paste and Al paste.

**Figure 5 materials-14-00765-f005:**
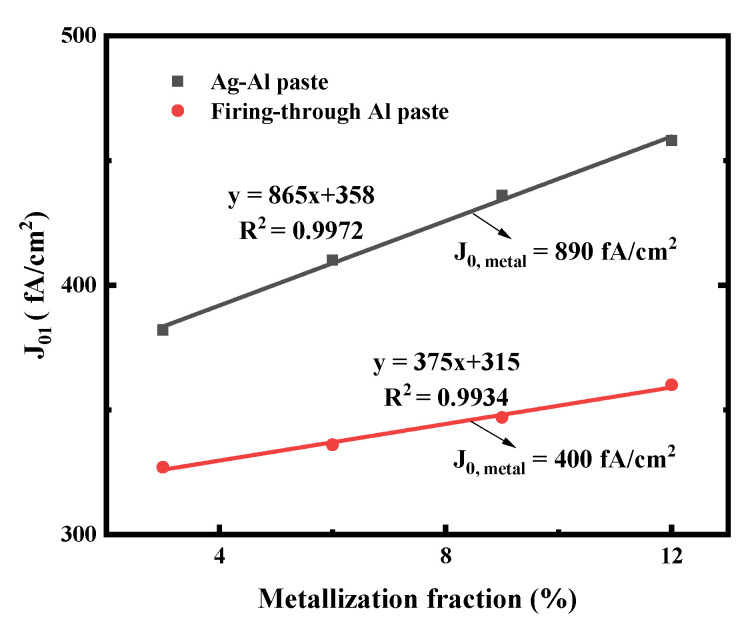
J_01_ extracted from Suns-V_OC_ as a function of metal fraction variation on the cell boron emitter. The J_0, metal_ is then extracted from the linear fit (solid lines).

**Figure 6 materials-14-00765-f006:**
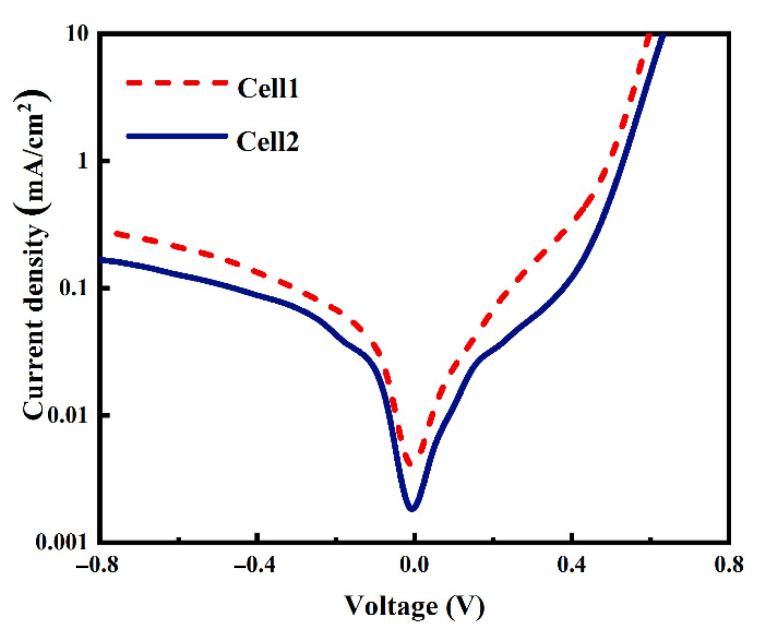
Dark J-V curves of the two cells using Ag-Al paste and firing-through Al paste.

**Figure 7 materials-14-00765-f007:**
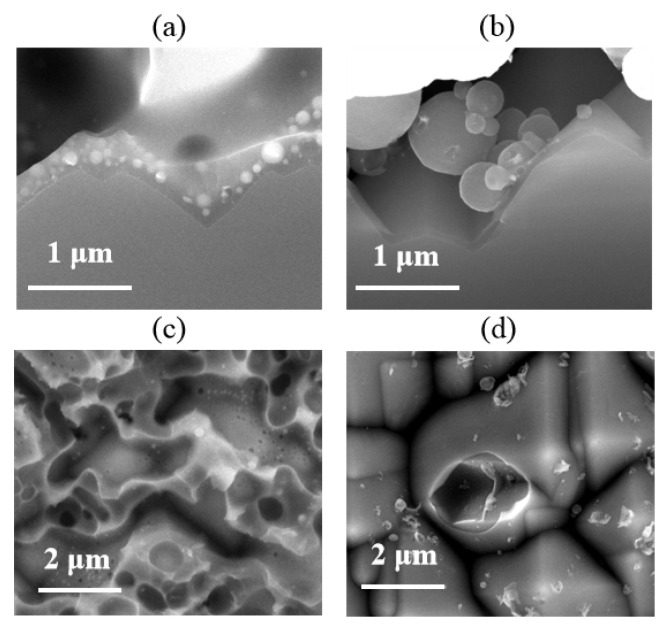
(**a**,**b**) Cross section scanning electron microscope (SEM) images of Cell 1 and Cell 2, obtained at the Si/metallization interface; (**c**,**d**) SEM images of Si substrate surface etched area after the paste removal.

**Figure 8 materials-14-00765-f008:**
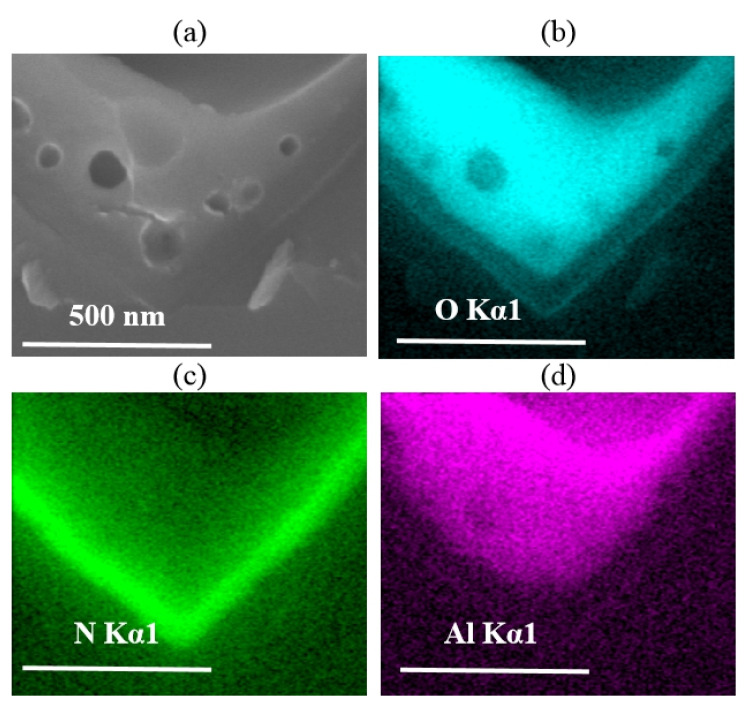
(**a**) Si/metallization interface SEM images of Cell 2; (**b**–**d**) Energy dispersive spectroscopy (EDS) mapping images obtained at Si/metallization contact area shown in (**a**). Different element such as O, N, Al are identified as different colors.

**Figure 9 materials-14-00765-f009:**
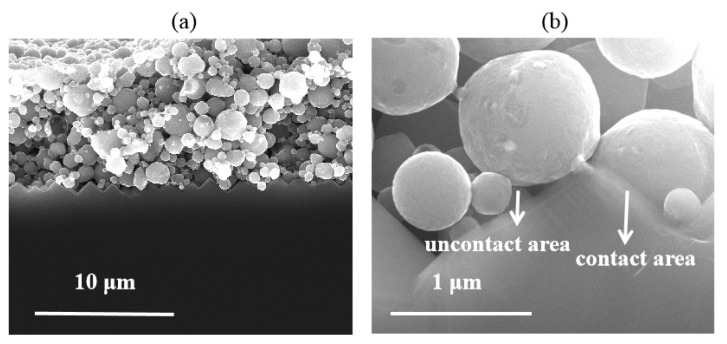
Cross-section SEM images obtained at the metallization/Si contact area of Cell 2: (**a**) low-magnification images; (**b**) high-magnification images. The SEM images show a discontinuous contact between the firing-through Al paste and Si substrate of Cell 2.

**Figure 10 materials-14-00765-f010:**
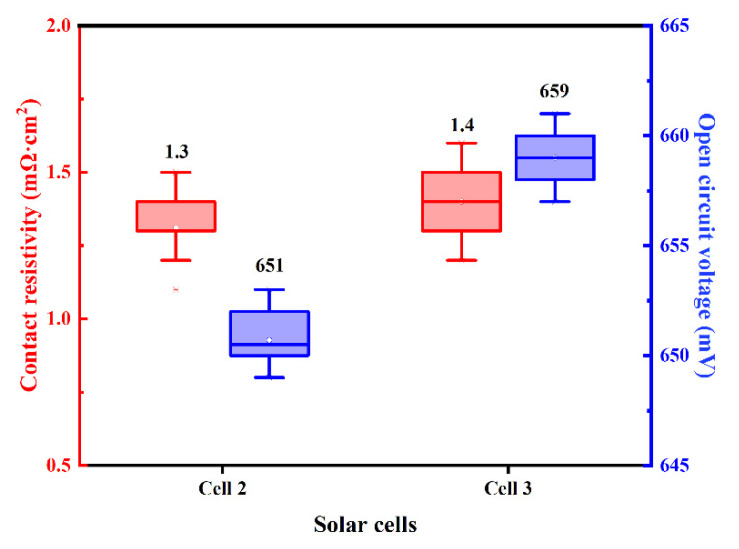
The distribution of the ρ_c_ and V_OC_ of the solar cells using the firing-through Al paste without and with Si particles.

**Figure 11 materials-14-00765-f011:**
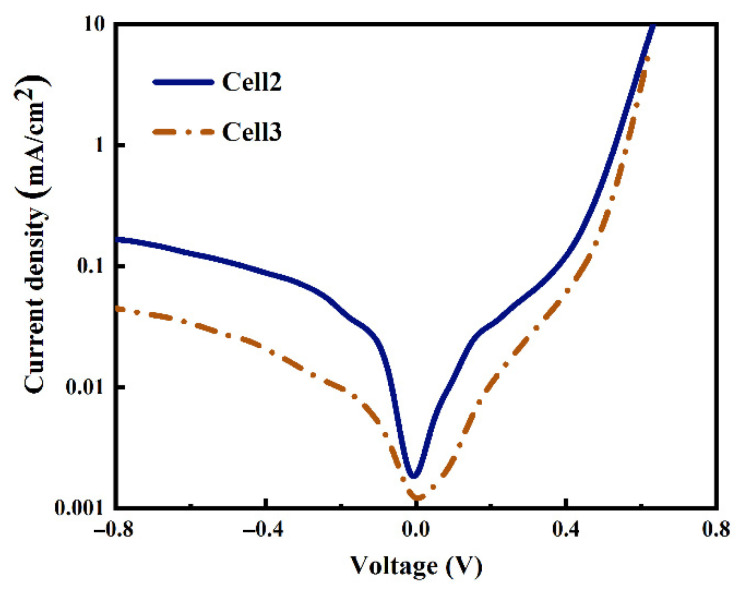
Dark J-V curves of the two cells using firing-through Al paste with and without adding silicon particles.

**Figure 12 materials-14-00765-f012:**
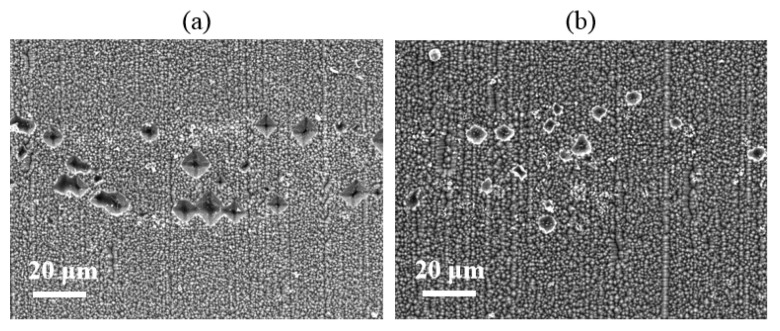
Si substrate surface etched area scanning electron microscope (SEM) images of the cells with (**a**) and without (**b**) adding Si particles.

**Figure 13 materials-14-00765-f013:**
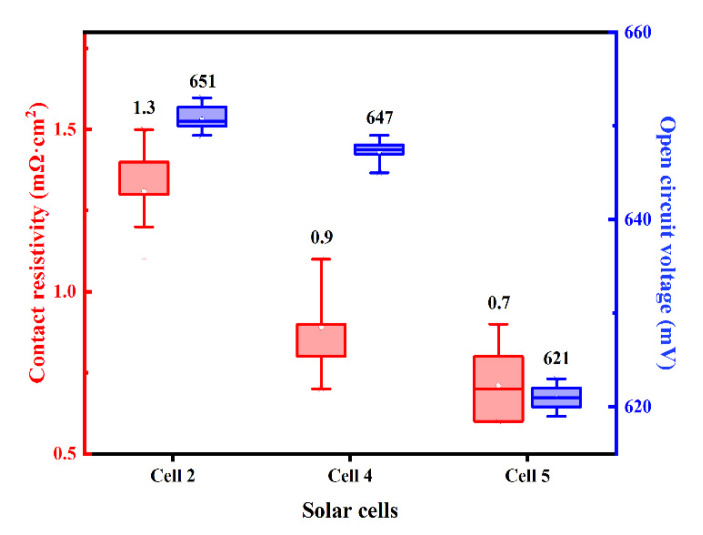
The distribution of the ρ_c_ and V_OC_ of the solar cells using the firing-through Al paste within different particle size Al powders.

**Figure 14 materials-14-00765-f014:**
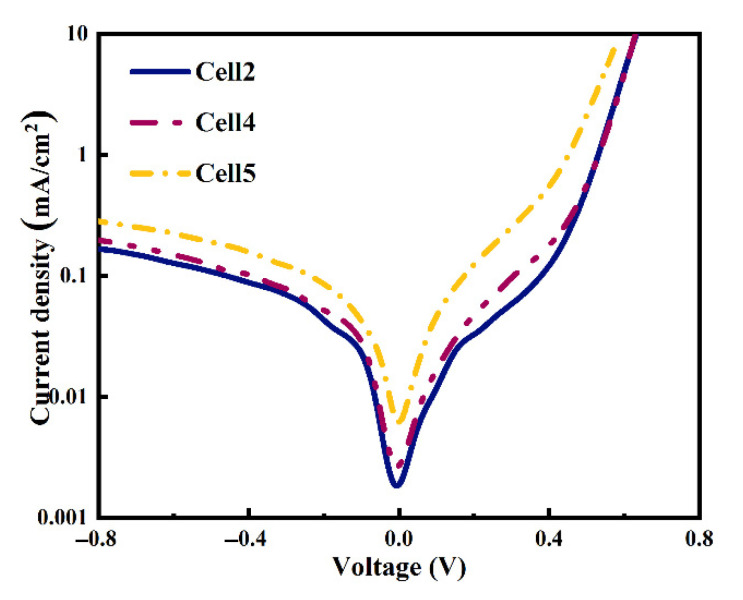
Dark J-V cures of the cells using different particle size Al powders.

**Figure 15 materials-14-00765-f015:**
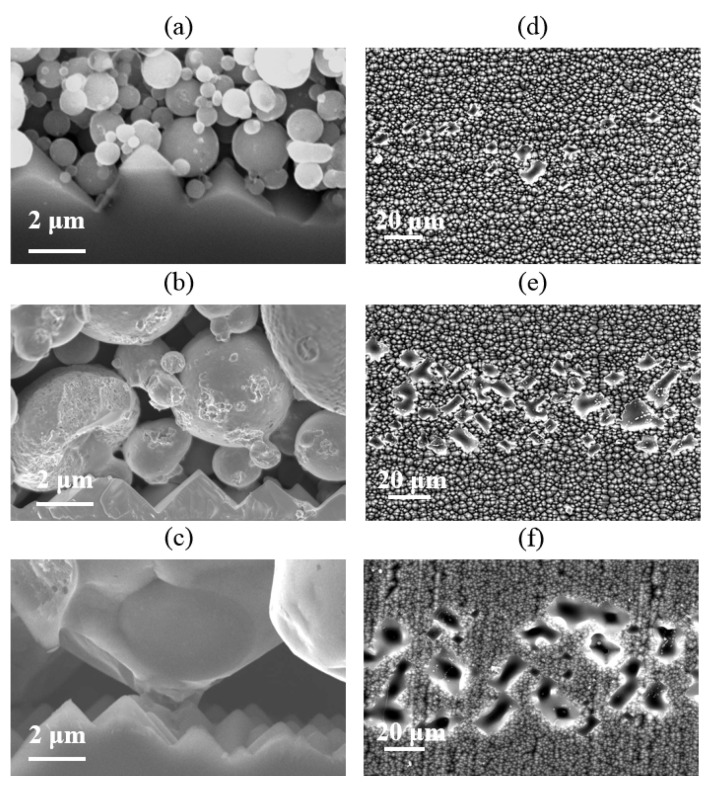
(**a**–**c**) Cross section SEM images; (**d**–**f**) Si substrate surface etched area after the paste’s removal.

**Figure 16 materials-14-00765-f016:**
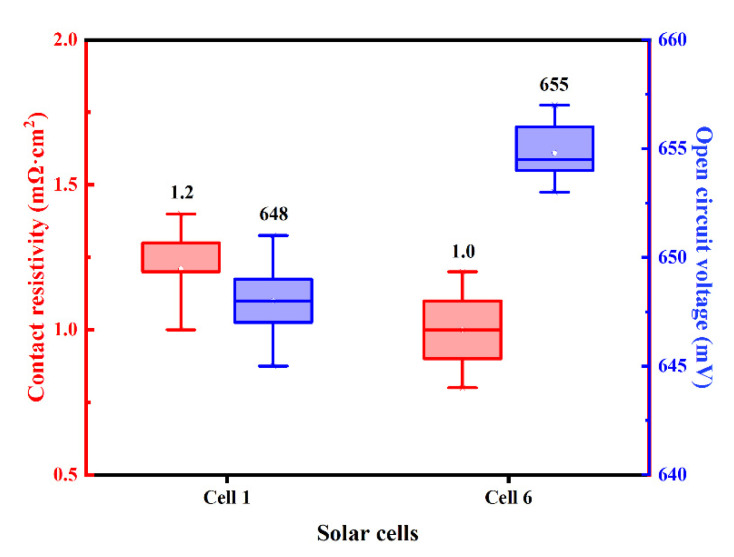
The experimental electrical data distribution of the cells fabricated with Ag-Al paste and improved firing-through Al paste.

**Figure 17 materials-14-00765-f017:**
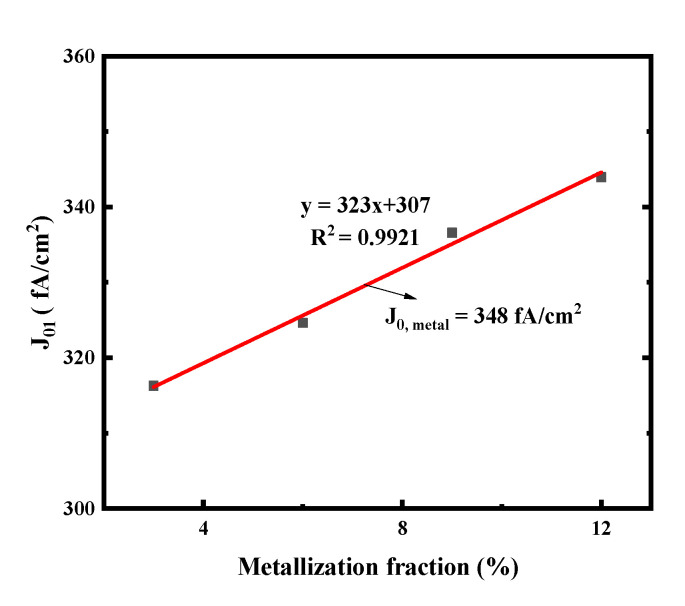
J_01_ extracted from Suns-V_OC_ as a function of metal fraction variation on the cell boron emitter. The J_0, metal_ is then extracted from the linear fit (solid lines).

**Figure 18 materials-14-00765-f018:**
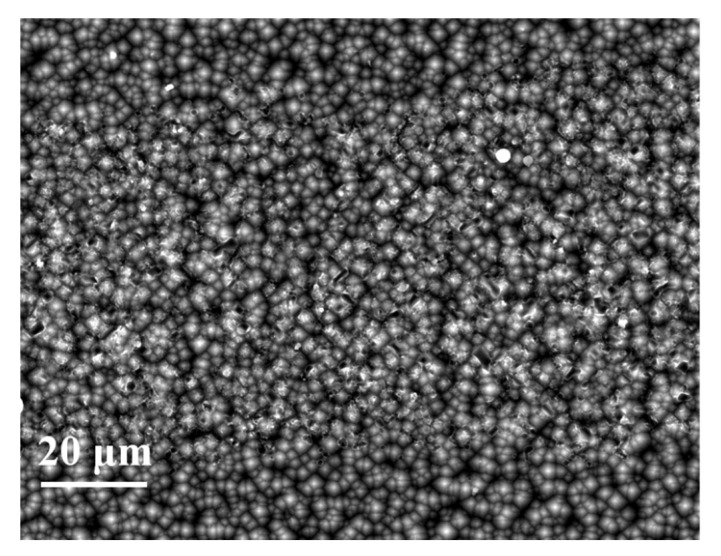
Si substrate surface etched area SEM images obtained at the Si/metallization interface of Cell 6 after the paste removal.

**Table 1 materials-14-00765-t001:** The electrical characteristics of the solar cells under 1 sun light illumination.

Solar Cells	V_OC_ (mV)	J_SC_ (mA/cm^2^)	FF (%)	Eff (%)	R_S_ (Ω)
Cell 1	648	40.15	79.76	20.88	0.0017
Cell 6	655	39.59	76.06	19.71	0.0034

**Table 2 materials-14-00765-t002:** The metal recombination, contact resistivity and implied electrical characteristics of the solar cells when use Ag-Al paste and improved firing-through Al paste.

Solar Cells	i-V_OC_ (mV)	pFF (%)	pEff (%)	J_01_ (fA/cm^2^)
Cell 1	648	82.73	21.00	382
Cell 6	655	83.48	21.34	347

## Data Availability

Data is contained within the article.

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
