# Peer review of "Low Recombination Firing-Through Al Paste for N-Type Solar Cell with Boron Emitter"

_materials, 2021, doi:10.3390/ma14040765_

Round 1
Reviewer 1 Report
The article entitled “Low Recombination Firing-through Al Paste for N-type Solar Cell Boron Emitter” by Peng Zhu et al. is well structured and is in line with the aims of the journal Materials. Before it can be accepted for publication some minor issues should be addressed, in particular:
- Abstract: please revise the sentence “Compared with the traditional process by using Ag or Ag-Al paste, the new strategy that in this work was much cheaper and more convenient, which could be an improved application in Al pastes.”
- Materials and Methods section is poorly detailed. For example: the commercial silver (Ag) paste was provided by?; the screen printer used in this study was?
- Please revise the sentences “Based on the above, the continuous Ag nano colloids on the Si substrate could attribute to the high J0, metal data of the cell using Ag-Al paste” and “Based on the above, we speculate that the lower reverse leakage of the cell 3 attribute to the addition of silicon particles reducing the defects to the cell caused by the paste.”
Author Response
Dear Editor,
We have studied the valuable comments from you, the assistant editor and reviews carefully, and tried our best to revise the manuscript. The point to point responds to the reviewer’s comments are listed as following:
Response to Reviewer 1 Comments
Point 1: Abstract: please revise the sentence “Compared with the traditional process by using Ag or Ag-Al paste, the new strategy that in this work was much cheaper and more convenient, which could be an improved application in Al pastes.”
Response 1: Thank you for your valuable advice. This sentence has been revised to “The utilization of the new kind of Al paste in this work was much cheaper and more convenient, compared with the traditional process by using Ag or Ag-Al paste.”.
Point 2: Materials and Methods section is poorly detailed. For example: the commercial silver (Ag) paste was provided by?; the screen printer used in this study was?
Response 2: Thank you for your valuable suggestion. The detailed materials and methods have been added in the manuscript.
Point 3: Please revise the sentences “Based on the above, the continuous Ag nano colloids on the Si substrate could attribute to the high J0, metal data of the cell using Ag-Al paste” and “Based on the above, we speculate that the lower reverse leakage of the cell 3 attribute to the addition of silicon particles reducing the defects to the cell caused by the paste.”
Response 3: Thank you for your valuable suggestion. These sentences have been revised in the manuscript.
We have tried our best to improve the manuscript. We appreciate for Editors/Reviewers’ warm work earnestly, and hope that the correction will meet with approval.
Once again, thank you very much for your comments and suggestions.

Reviewer 2 Report
Paper Low Recombination Firing-through Al Paste for N-type Solar Cell Boron Emitter deals with different metal pastes used for fabrication of front N-PERT electrode. Paper shows progress in investigations of different paste components and their influence on solar cell parameters. However, before the publication some issues has to carefully addressed:
- Title - I wold suggest to add word with :Low Recombination Firing-through Al Paste for N-type Solar Cell with Boron Emitter
- why authors used term of pseudo in the case of conversion efficiency and open circuit voltage? (line 23, 336, 381)
- missing space - line 105 - 120nm
- I suggest to replace Scheme 1 with Figure 1
- Scheme 1b - I would recommend to prepare a legend when different colour is related with different layer instead of using arrows and text - it is illegible.
- please rotate by 180 deg the legend of Open circuit voltage in Fig. 3, 9, 12 and 15
- please use the same legend/nomenclature for sample Cell 1 and Cell 2 in Fig 4 and 5
- please provide detailed analysis of dark J-V curves shown in Fig. 5, 10, 13. Why authors do not use ideality factor and series and shunt resistances to compare solar cells with different front metallization?
- Part 3.3 - are smaller Al particles available? Authors only investigate paste with bigger Al particles and obtain obvious results concerning open circuit voltage.
- Part 3.3 - Why authors do not discuss the Al particle size influence on contact resistivity?
- Part 3.3, lines 304-305. Authors suppose that paste with the biggest Al particles (Cell 5) during firing may destroy the pn junction. Why the contact resistance obtained for this cell is relatively low?
- Part 3.4 - What is a reason that author do not show light J-V characteristics of Cell1 and Cell6?
- line 362 - some writing error in Fig. 17 description: Si substrate surface etched area SEM images obtained at t655he Si/metallization interface of Cell 6 after the paste removal.
Author Response
Dear Editor,
We have studied the valuable comments from you, the assistant editor and reviews carefully, and tried our best to revise the manuscript. The point to point responds to the reviewer’s comments are listed as following:
Response to Reviewer 2 Comments
Point 1: Title - I wold suggest to add word with :Low Recombination Firing-through Al Paste for N-type Solar Cell with Boron Emitter.
Response 1: Thank you for your valuable advice. The title has been modified to “Low Recombination Firing-through Al Paste for N-type Solar Cell with Boron Emitter”.
Point 2: why authors used term of pseudo in the case of conversion efficiency and open circuit voltage? (line 23, 336, 381)
Response 2: Thank you for your question. In this work, the front busbars (main grid) of solar cells we prepared in this work was also using Al paste. The high series resistances of solar cells are shown in the electrical property because of the high line resistance of Al, which could not be comparable to that of commercial solar cells. On the other hand, further work to improve the series resistance of solar cell is in progress. We will try our best to optimize the metallization research of silicon solar cell. Thank you for your precious advice again.
Point 3: missing space - line 105 - 120nm
Response 3: Thank you for your valuable advice. The mistake has been corrected in the manuscript.
Point 4: I suggest to replace Scheme 1 with Figure 1
Response 4: Thank you for your valuable advice. The Scheme 1 has been amended to Figure 1 in the manuscript.
Point 5: Scheme 1b - I would recommend to prepare a legend when different colour is related with different layer instead of using arrows and text - it is illegible.
Response 5: Thank you for your valuable advice. The legends related with different layer have been added in Figure 1.
Point 6: please rotate by 180 deg the legend of Open circuit voltage in Fig. 3, 9, 12 and 15.
Response 6: Thank you for your valuable advice. The manuscript has been revised.
Point 7: please use the same legend/nomenclature for sample Cell 1 and Cell 2 in Fig 4 and 5.
Response 7: Thank you for your valuable suggestion. The manuscript has been revised.
Point 8: please provide detailed analysis of dark J-V curves shown in Fig. 5, 10, 13. Why authors do not use ideality factor and series and shunt resistances to compare solar cells with different front metallization?
Response 8: Thank you for your valuable suggestion. The detailed analysis of dark J-V curves shown in Fig. 5, 10, 13 have been provided in the revised manuscript. The metallization paste used in front busbars (main grid) of solar cells we prepared in this work was the same as that of fine grid. The high series resistances of solar cells are shown in the electrical property because of the high line resistance of Al.
Point 9: Part 3.3 - are smaller Al particles available? Authors only investigate paste with bigger Al particles and obtain obvious results concerning open circuit voltage.
Response 9: Thank you for your question. When the size of Al particle is smaller, the stability of Al powder will get worse. The smallest size of Al particles used in this work is 1~3 μm. The effect of different sizes of Al particle was compared in the manuscript in details. Actually, further work using nanoscale Al particle in the metallization of silicon solar cell is another consideration. The recommendation of reviewer is excellent.
Point 10: Part 3.3 - Why authors do not discuss the Al particle size influence on contact resistivity?
Response 10: Thank you for your question. The discussion of the Al particle size influence on contact resistivity has been added in the revised manuscript.
Point 11: Part 3.3, lines 304-305. Authors suppose that paste with the biggest Al particles (Cell 5) during firing may destroy the pn junction. Why the contact resistance obtained for this cell is relatively low?
Response 11: Thank you for your question. In this work, the morphology and oxygen content of Al particles have significant effect on the surface metal recombination and the contact resistivity. When the small-size Al particles with high oxygen content are applied in the metallization paste, the physical contact area shows smaller than that of large-size Al particles, which is along with the less serious damage of the substrate. Subsequently, the utilization of small-size Al particles showed a lower Voc value and a higher contact resistance. On the other hand, the large-size Al particle is prone to damage the pn junction, which would course defects of cells shown as Fig. 14 and the SEM images in Fig. 15.
Point 12: Part 3.4 - What is a reason that author do not show light J-V characteristics of Cell1 and Cell6?
Response 12: Thank you for your question. The front busbars (main grid) of solar cells we prepared in this work was also using Al paste. The high series resistances of solar cells are shown in the electrical property because of the high line resistance of Al. The light J-V characteristics of solar cells could not be comparable to that of commercial solar cells.
Point 13: line 362 - some writing error in Fig. 17 description: Si substrate surface etched area SEM images obtained at t655he Si/metallization interface of Cell 6 after the paste removal.
Response 13: Thank you for your valuable suggestion. The mistake has been corrected in the manuscript.
We have tried our best to improve the manuscript. English language and style are carefully checked. We appreciate for Editors/Reviewers’ warm work earnestly, and hope that the correction will meet with approval.
Once again, thank you very much for your comments and suggestions.

Round 2
Reviewer 2 Report
The corrected paper is publishable.